# Nuclear Medicine and Radiological Imaging of Pancreatic Neuroendocrine Neoplasms: A Multidisciplinary Update

**DOI:** 10.3390/jcm11226836

**Published:** 2022-11-18

**Authors:** Daniela Prosperi, Guido Gentiloni Silveri, Francesco Panzuto, Antongiulio Faggiano, Vincenzo Marcello Russo, Damiano Caruso, Michela Polici, Chiara Lauri, Angelina Filice, Andrea Laghi, Alberto Signore

**Affiliations:** 1Nuclear Medicine Unit, Department of Medical-Surgical Sciences and of Translational Medicine, Sant’Andrea University Hospital, Sapienza University of Rome, 00189 Roma, Italy; 2Digestive Disease Unit, Department of Medical-Surgical Sciences and Translational Medicine, Sant’Andrea University Hospital, ENETS Center of Excellence, Sapienza University of Rome, 00189 Roma, Italy; 3Endocrinology Unit, Department of Clinical and Molecular Medicine, Sant’Andrea University Hospital, ENETS Center of Excellence, Sapienza University of Rome, 00189 Roma, Italy; 4Radiology Unit, Department of Medical Surgical Sciences and Translational Medicine, Sant’Andrea University Hospital, Sapienza University of Rome, 00189 Roma, Italy; 5Nucler Medicine Unit, AUSL-IRCCS di Reggio Emilia, 42123 Reggio Emilia, Italy

**Keywords:** panNEN, pancreatic, neuroendocrine, PET/CT, ^68^Ga-SSA, ^18^F-FDG, ^68^Ga-exendin-4, ^18^F-DOPA, radiomics

## Abstract

Pancreatic neuroendocrine neoplasms (panNENs) are part of a large family of tumors arising from the neuroendocrine system. PanNENs show low–intermediate tumor grade and generally high somatostatin receptor (SSTR) expression. Therefore, panNENs benefit from functional imaging with ^68^Ga-somatostatin analogues (SSA) for diagnosis, staging, and treatment choice in parallel with morphological imaging. This narrative review aims to present conventional imaging techniques and new perspectives in the management of panNENs, providing the clinicians with useful insight for clinical practice. The ^68^Ga-SSA PET/CT is the most widely used in panNENs, not only fr diagnosis and staging purpose but also to characterize the biology of the tumor and its responsiveness to SSAs. On the contrary, the ^18^F-Fluordeoxiglucose (FDG) PET/CT is not employed systematically in all panNEN patients, being generally preferred in G2–G3, to predict aggressiveness and progression rate. The combination of ^68^Ga-SSA PET/CT and ^18^F-FDG PET/CT can finally suggest the best therapeutic strategy. Other radiopharmaceuticals are ^68^Ga-exendin-4 in case of insulinomas and ^18^F-dopamine (DOPA), which can be helpful in SSTR-negative tumors. New promising but still-under-investigation radiopharmaceuticals include radiolabeled SSTR antagonists and ^18^F-SSAs. Conventional imaging includes contrast enhanced CT and multiparametric MRI. There are now enriched by radiomics, a new non-invasive imaging approach, very promising to early predict tumor response or progression.

## 1. Introduction

The neuroendocrine neoplasms (NENs) are heterogeneous diseases in terms of pathological, molecular, and clinical features. Although they are still considered rare cancers, their incidence has dramatically increased over the last decades, likely due to the improved accuracy of diagnostic tools [1]. Their clinical outcome is extremely variable, depending on several factors, including primary tumor site, grading, and staging, which may be combined to assess tumor prognosis [2]. Among these features, tumor grading is widely accepted as the strongest prognostic factor, able to predict response to medical treatment, risk of recurrence after radical surgery, and overall survival. As proposed by the WHO classification, four categories of NENs are identified depending on the grading system: neuroendocrine tumors (NET) G1 (well-differentiated morphology and Ki67 < 3%), NET G2 (well-differentiated morphology and Ki67 3–20%), NET G3 (well-differentiated morphology and Ki67 > 20%), and neuroendocrine carcinomas (NEC) G3 (poorly differentiated morphology and Ki67 20%) [3].

According to the primary tumor site, two major groups of NENs may be distinguished— gastrointestinal NENs and pancreatic NENs (panNENs)—which should be separately managed due to their different risk profiles and clinical behaviors [4].

The diagnostic work-up of NENs is based on cross-sectional radiological procedures (computed tomography and magnetic resonance imaging) and nuclear medicine examinations, including ^68^Gallium-somatostatin-analogues (^68^Ga-SSA) positron emission tomography/computed tomography (PET/CT) and, in selected patients, ^18^F-fluorodeoxyglucose (^18^F-FDG PET/CT). More recently, the application of radiomics has been suggested as a promising diagnostic approach able to predict tumor behavior and patients’ clinical outcomes [5].

## 2. Conventional Nuclear Medicine Radiopharmaceuticals

### 2.1. 68Ga-SSA PET/CT

Over the years, nuclear medicine has played a central role in the diagnosis of NENs. The identification of somatostatin (SST) in 1973, and subsequently, the discovery of the five somatostatin receptor subtypes (SSTR1 to SSTR5) in the early 1990s, has provided a relevant diagnostic and therapeutic opportunity. Somatostatin receptors (SSTRs) are widely distributed in healthy tissues, with distinct expression throughout the body, but interestingly, the expression of SSTRs is significantly enhanced in many human solid tumors where they are expressed alone or in various combinations, especially in gastro-entero-pancreatic neuroendocrine tumors (GEP-NENs) [6,7,8,9,10].

The use of native SST is hindered by its poor in vivo stability, and this has been a limiting factor for possible clinical applications. This problem has been competently addressed by the advent of synthetic SSA with better metabolic properties [11,12,13,14]. Somatostatin receptor scintigraphy (SRS) with ^111^In-pentetreotide or OctreoScan was the first peptide-based radiopharmaceutical that has been approved by the FDA (Octreoscan^®^ marketed in 1994). The internalization and retention in the cytoplasm of the radioligand-receptor complex is the rationale for SST imaging. Although SRS has been widely used in NENs diagnosis, it has many limitations, such as less favorable tumor/noise intensity ratio, especially in the abdomen (liver and bowel), low spatial resolution, moderate affinity for receptors, and, finally, high γ energy, which results in a high dose of radioactivity received by the patient. For these reasons, research has focused on other radioelements, and SRS has been replaced by novel radiopharmaceuticals. In particular, an important step forward was made thanks to the advent of the next generation of SSA labeled with the positron-emitter radiometal ^68^Ga, developed for clinical use with PET/CT [10,15].

In clinical practice, three ^68^Ga-labeled radiopharmaceuticals for PET/CT imaging are available: ^68^Ga-DOTA-Phe1-Tyr3-Octreotide (DOTA-TOC), ^68^Ga-DOTA-NaI3-Octreotide (DOTA-NOC), and ^68^Ga-DOTA-Tyr3-Octreotate (DOTA-TATE) [16]. These three radiopharmaceuticals have slightly different pharmacokinetic properties, mainly due to their different affinities for SSTRs subtypes. [10]. ^68^Ga-DOTA-TOC shows high affinity for SSTR2 and moderate for SSTR5, ^68^Ga-DOTA-TATE is specific to SSTR2, and ^68^Ga-DOTA-NOC binds with great affinity to SSTR2, SSTR3, and SSTR5 [17,18,19].

Despite the different receptor affinities, comparative studies showed no clinically significant difference among different radiopharmaceuticals [20,21,22]. Several studies have shown higher accuracy of ^68^Ga-SSA PET/CT for detection NETs, as compared with both SRS and conventional imaging [22,23,24,25,26]. Current guidelines recommend performing ^68^Ga-SSA PET/CT in NENs for diagnosis and staging, restaging, follow-up, prognostic evaluation, therapy decision-making, and for therapy monitoring [22,27]. Moreover, imaging with ^68^Ga-SSA PET/CT has the highest sensitivity (86–100%) and specificity (79–100%) for localizing panNENs, as well as for other NENs [28,29,30,31,32]. Its use has been shown to change the management (surgical, medical, and staging) in 20–55% of patients [30,31]. Nevertheless, the performance of ^68^Ga-SSA PET/CT in detecting insulinomas may be lower than other NENs, due to the relatively lower expression of SSTRs (25–30% of positivity) [32,33], and, even if the sensitivity of ^68^Ga-SSA PET/CT for insulinomas is higher compared with SRS (85% vs. 50–60%), its diagnosis remains challenging [34,35]. Although insulinomas are generally benign conditions, approximately 5–15% of patients show a more aggressive behavior. Malignant insulinomas are characterized by distant metastases, lymph node involvement and direct invasion into surrounding peri-pancreatic tissue, or presence of lymphatic and vascular invasion. Nevertheless, differently from benign insulinomas, most malignant insulinomas express SSTRs [36], thus representing a potential candidate for radioligand therapy [35,37].

The knowledge of the normal biodistribution of the radiopharmaceutical is imperative for optimal patient outcome. In particular, the interpretation of pancreatic findings requires caution, as this gland can show a variable degree of physiological/benign uptake, both with a diffuse and focal character (physiologic uptake in the pancreatic uncinate process, inflammation, intrapancreatic accessory spleen, and hyperplasia). Therefore, these findings must be correctly interpreted in light of conventional imaging methods, such as CT or MRI, in order to achieve an accurate diagnosis and to accurately differentiate a physiological from a pathological uptake. In particular, the uncinate process of the pancreas has a high density of SSTR2, 3, and 5, which explains the high activity that is frequently observed in this region (Figure 1) [38]. Indeed, as reported by several studies, increased pancreatic uptake on the head or uncinate process may be observed with variable frequency due to the great concentration of pancreatic polypeptide cells. This finding can be observed in more than one-third of patients, thus representing a potential source of misinterpretation, since the pancreas is one of the most frequent NENs primary sites [39,40]. In a recent study, the authors found that the uncinate process uptake can be observed in almost half of patients, therefore, in a percentage higher than that reported in previous studies, mainly presenting with a diffuse pattern. The authors also found that the uptake pattern can vary over time in the same patient in approximately one-third of cases [41].

An uncommon cause of false positive cases on ^68^Ga-SSA PET/CT is an intrapancreatic accessory spleen (Figure 2). In autoptic studies, the accessory spleen has an incidence of 10% and intrapancreatic accessory spleen of 2% [42], but clinical incidence is growing, likely related to the improvement of diagnostic imaging accuracy [43]. Accessory spleen can be found in splenic hilum (80%), pancreatic tail (20%), stomach, bowel, and genitals [44,45,46]. In the presence of an incidental and asymptomatic lesion to the pancreatic tail, a careful evaluation of the radiological images is crucial, and a combination of CT, MRI, nuclear medicine examinations, EUS-FNA, or EUS-guided core biopsy could be necessary for the diagnosis [44]. Moreover, the motion artifact, frequent in this abdominal region, can contribute to misinterpretation of pancreatic findings [47]. The pancreatic uptake should, therefore, be evaluated with caution, and, in this context, the comparison with morphological information, especially if performed with intravenous administration of contrast agent, has a crucial role. Despite the difficulties in interpreting some findings, the advent of ^68^Ga-SSA PET/CT has contributed to a significant improvement in the diagnosis of NENs so that currently, ^68^Ga-SSA PET/CT represents the gold-standard imaging modality for NENs [22,27] and is considered the method of choice to fully stage and localize the extent of disease in patients with non-insulinoma panNENs [48].

### 2.2. 18F-FDG PET/CT

^18^F-FDG is a glucose analogue that concentrates in neoplastic cells in proportion to their glucose metabolic activity and represents the main radiopharmaceutical used in PET imaging, especially for oncological studies [49]. In the management of NENs, ^18^F-FDG PET/CT has been suggested as an alternative tool, although it is still an object of active debate, in particular for the assessment of lower-grade tumors (G1 and low-grade G2 Ki-67 < 10%).

Indeed, most panNENs are well-differentiated lesions that express SSTRs on their surface, are characterized by low Ki-67 and proliferation index, and usually do not use the metabolic pathway. However, during the disease’s progression, they may lose the ability to express SSTRs and become more aggressive, thus increasing their glucose metabolism [50].

Guidelines (both form the European Association of Nuclear Medicine (EANM) and the European Neuro-Endocrine Tumor Society (ENETS)) [22,27,51] suggest the use of ^18^F-FDG PET/CT in G3 NET, NEC, and higher-grade G2 (e.g., Ki-67 10–20%) NEN, which generally show higher glucose metabolism and ^18^F-FDG avidity. The European Society for Medical Oncology (ESMO) guidelines, published in 2020 [52], state that optimal diagnostic and prognostic information can be achieved by directing all NEN G2–G3 patients to PET/CT with both ^18^F-FDG and ^68^Ga-SSA PET/CT; however, the real utility of such unconfirmed approach in routine practice is still debated.

There is emerging evidence that the increase of glucose metabolism correlates with the presence of more aggressive tumor cell clones and indicates worse prognosis [53]. In addition to visual assessment of ^18^F-FDG distribution, some semi-quantitative parameters can be also evaluated. The most commonly used in clinical practice is represented by maximum and mean standardized uptake value (SUVmax and SUVmean, respectively), but, more recently, metabolic tumor volume (MTV) and total lesion glycolysis (TLG) are also emerging as valuable tools in several clinical settings. Nevertheless, their prognostic value in NENs has not yet been validated [53,54]. Several authors have investigated the potential clinical role of ^18^F-FDG PET/CT in low-grade patients, demonstrating that it can provide useful prognostic information, even in this setting, in order to improve patient clinical management [27,53,54,55]. However, sensitivity of ^18^F-FDG PET/CT in GEP-NET G1–G2 ranges between 40% and 60%, while it increases up to almost 95% in G3 tumors [55,56].

Recently, Magi et al. [57] evaluated the prognostic value of ^18^F-FDG PET/CT in 55 patients with G1 GEP-NETs including 24 with pancreatic and 31 with gastrointestinal NENs. Approximately half of GEP-NETs G1 showed positive findings, more evident in patients with pancreatic tumors. Positive results at ^18^F-FDG PET/CT correlate with a worse prognosis and can be useful in selecting high-risk patients, who may benefit from more aggressive treatments. Therefore, an ^18^F-FDG scan can be useful for prognostic purposes also in well-differentiated G1 and low-grade G2 (Ki-67 < 10%) tumors.

In the last few years, several papers have investigated the combined use of ^18^F-FDG and ^68^Ga-SSA PET/CT in the management of NENs, generally in intermediate- or high-grade panNETs, to obtain useful information on tumor heterogeneity, the characterization of SSTR expression, and tumor grade (Figure 3).

Using a combined approach, two different aspects of tumor biology may be explored: ^68^Ga-SSA PET/CT can provide information about the expression of SSTRs, thus evaluating the grade of tumor differentiation and guiding to targeted therapies. Conversely, ^18^F-FDG PET/CT can assess the prognosis and tumor aggressiveness, both in the staging and in restaging, thus providing useful information for risk stratification. Indeed, when increased tumor aggressiveness is suspected in patients with well-differentiated metastatic NETS, ^18^F-FDG PET/CT scans should be always performed.

Cingarlini et al. [58] evaluated the role of combined ^68^Ga-DOTA-TOC and ^18^F-FDG PET/CT in the diagnostic work-up of panNENs. ^18^F-FDG scan results were associated with tumor progression and death risk in this setting of patients.

In a recent study, Paiella et al. [59] performed combined imaging using either ^18^F-FDG and ^68^Ga-DOTA-TOC scan in 124 patients with G1–G2 non-advanced unifocal resected panNETs. Authors found that the dual radiopharmaceutical approach may be used in identifying the disease (combined sensitivity of 99%) to predict the presence of possible recurrence and to give crucial prognostic information. In particular, ^18^F-FDG positivity was associated with larger tumors’ size and with higher Ki-67 values.

An interesting study [60] investigated the complementary use of ^68^Ga-DOTA-TATE and ^18^F-FDG PET/CT scans in the evaluation of low- and intermediate- versus high-grade NENs. Patients were divided into a control group, which included well-differentiated cases (Ki-67 < 20%, G1–G2), and an experimental group, consisting of poorly differentiated cases (Ki-67 > 20%, G3). All patients with high-grade and poorly differentiated NETs were positive on both ^68^Ga-DOTA-TATE and ^18^F-FDG PET/CT scans; in particular, both imaging methods were quite similar in identifying multiple lesions (>10) or distant metastases. Authors also suggested the use of both PET examinations to select the patients who could benefit from a peptide receptor radionuclide therapy (PRRT). On the basis of several systematic reviews [61,62], this combined approach should be adopted either at basal time in the case of heterogeneity between different lesions or within the same lesion to the expression of the SSTRs, or in patients showing a discrepancy between conventional radiological imaging and ^68^GaSSA PET/CT, at the time of the first diagnosis, or during the follow-up after changing treatment and before starting other line of therapy.

Chan et al. [63] proposed a new grading scheme for metastatic NEN, called the “NETPET Score”, by using both dual-radiopharmaceutical PET scan. This score identifies five categories of patients: P0, with both negative ^68^Ga-SSA PET/CT and ^18^F-FDG PET/CT; P1, with positive SRS-PET/CT only; P2–P4, intermediate cases with both positive scans; and P5, with positive ^18^F-FDG PET/CT only. The “NETPET Score” correlates significantly with tumor grade and overall survival, provides prognostic information, and helps to better select patients who can benefit from PRRT. Several studies demonstrated that ^18^F-FDG PET/CT plays an important role for patient selection for PRRT and for therapy prediction of response, although it is not part of the standard pre-PRRT protocol; in particular, ^18^F-FDG positivity does not exclude the patients who are candidates for the PRRT.

Thapa et al. [64] analyzed combined imaging using either ^99m^Tc-Hynic-TOC or ^68^Ga-DOTA-TATE scans with ^18^F-FDG in 50 patients with metastatic GEP-NET prior and after PRRT therapy with 177-Lutetium (^177^Lu)-DOTA-TATE. They found that ^18^F-FDG positivity, both in high- and low-grade NEN, was correlated with a worse response and poorer prognosis to PRRT.

Zhang et al. retrospectively enrolled 495 patients with metastatic NENs, including 199 patients with panNEN, who were treated with PRRT with a long-term follow-up [65]. All patients were studied with both ^68^Ga-DOTA-TOC/TATE/NOC and ^18^F-FDG PET/CT prior to treatment with ^177^Lu or 90Yttrium (^90^Y)-DOTA-TOC/DOTATATE. In this setting, negative ^18^F-FDG PET/CT correlates with a better median OS and PFS compared with positive ^18^F-FDG PET/CT (median OS: 114.3 vs. 52.8 months and median PFS: 36.9 vs. 22.4 months, respectively; for both *p* < 0.001).

Sansovini et al. [66] evaluated the role of ^18^F-FDG PET/CT in 60 patients with advanced pancreatic neuroendocrine tumors and efficacy of ^177^Lu -DOTA-TATE even in this setting.

The authors found that ^18^F-FDG PET/CT is an independent prognostic factor and that the patients with a negative ^18^F-FDG PET/CT showed a significantly better outcome after PRRT therapy, compared with those with positive scan.

Alevroudis et al. [67] published a systematic review and a meta-analysis focusing on the clinical utility of ^18^F-FDG PET/CT imaging as a predictive tool for PRRT administration in patients with NENs. Although relatively favorable outcomes are evident in ^18^F-FDG PET/CT-positive patients with disease control rates as high as 74%, a higher risk for progression and death in this subset could be observed. Therefore, ^18^F-FDG PET/CT imaging could represent a crucial predictive tool for PRRT in patients with NENs. Dual-functional imaging should probably be considered before starting PRRT, both to delineate tumor SSTRs’ expression and glycolytic metabolic activity—aiming to plan a personalized treatment strategy (PRRT or other systemic treatments), for a prognostic disease evaluation with potential implications on the intensity of the surveillance strategy—and as a predictive tool. However, further well-designed randomized controlled trials, aiming to assess the clinical utility of ^18^F-FDG PET/CT imaging across a wide range of NENs and lower-grade tumors, are warranted.

In the recent EANM Focus 3 [68], consensus was reached on employing ^18^F-FDG PET/CT in NEC, G3 NET, and in G1–2 NET with mismatched lesions (CT-positive/-SSA PET/CT-negative). Furthermore, the use of ^18^F-FDG PET/CT in GEP-NEN patient candidates to PRRT was discussed, and its use for GEP-NET G2 patients was recommended by the panelists.

### 2.3. 68Ga-Exendin-4

As previously mentioned, insulinomas are particularly challenging conditions from both a diagnostic and clinical management point of view. They are characterized by hypoglycemia, which is often difficult to control clinically, making resection the only curative treatment. Conventional imaging studies can often be negative, especially in the presence of lesions. In this case, functional imaging can be crucial.

^68^Ga-SSA PET/CT has a lower sensitivity in detecting insulinomas other than pan-NENs, as SSTR expression is observed in only 25–30% of cases [33]. For this reason, new radiopharmaceuticals have recently been developed. Amongst them, agonists of glucagon-like peptide-1 receptor (GLP-1R) agonists showed higher sensitivity than SSTR2 analogues [48,69]. GLP-1R is a G-coupled receptor protein located in pancreatic beta cells, which plays a regulatory role in insulin secretion.

Exendin-4 has become the most widely accepted radiopharmaceutical thanks to its strong binding affinity with GLP-1R in vitro and its excellent resistance to serum degradation [69]. The two most commonly used radio compounds are ^68^Ga-DOTA-exendin-4 and ^68^Ga-NOTE-MAL-Cys39-exendin-4. PET/CT with these two radiopharmaceuticals allows accurate localization of the primary lesion thanks to the high tumor-to-background ratio and high spatial resolution [70,71]. PET imaging of the GLP-1 receptor has been shown to be superior to CT and MRI in detecting small insulinomas.

However, GLP-1R-based functional imaging has some pitfalls that can lead to a false negative test, in particular, for tumors closed to areas of physiologic uptake of the radiopharmaceutical, such as the pancreatic tail near the left kidney or near the pancreaticoduodenal region. Another limitation of this radiopharmaceutical is the low GLP-1R expression reported in malignant insulinomas [72,73]. Indeed, in a comparative study of 11 patients with insulinoma, only 36% of malignant insulinoma expressed GLP-1 receptors [36]. Exendin-4 has also been investigated for potential therapeutic use by labelling it with ^117^Lu for the treatment of patients with widespread disease [74]. However, ^117^Lu-labelled exendin-4 compounds have not been shown to be suitable for PRRT due to their high renal toxicity [75].

### 2.4. 18F-DOPA

PET/CT scan using 18-fluoride-dihydroxyphenyl-alanine (DOPA) (^18^F-DOPA) has also been used for diagnosis and staging of NENs. Despite its being used mostly as an alternative radiopharmaceutical to ^68^Ga-SSA, several studies demonstrate the high sensitivity and diagnostic accuracy of PET/CT with ^18^F-DOPA for the identification of well-differentiated NETs [76,77]. At the moment, the role of ^18^F-DOPA PET/CT in imaging entero-pancreatic NENs is under debate.

In a recent meta-analysis published by Piccardo et al. focused on 112 patients with small intestine NEN, the sensitivity of ^18^F-DOPA PET/CT and ^68^Ga-SSA PET/CT were, respectively, 83% and 88% on patient-based analysis, 89% and 92% on region-based analysis, and 95% and 82% on lesion-based analysis [78]. This means that there are a considerable number of lesions that do not show ^68^Ga-SSA uptake but can concentrate ^18^F-DOPA. Nevertheless, the clinical utility of ^18^F-DOPA in therapy decision-making is lower than ^68^Ga-SSA.

Understanding the importance of DOPA metabolic pathway and biodistribution is crucial for ensuring proper therapeutic application. Entero-pancreatic NENs incorporate certain types of amino acids such as DOPA and decarboxylate thanks to the aromatic L-amino acid decarboxylase. ^18^F-DOPA binds to the neutral amino acid transporter, and, after internalization (as DOPA), it is converted to ^18^F-4,5-DOPA dioxygenase extradiol (DODA) and stored in cytoplasmic vesicles. Increased uptake of ^18^F-DOPA by NETs is due mainly to the over-expression of the amino acid transport system [79].

The current clinical use of ^18^F-DOPA is focused on studying NEN variants with low or variable SSTRs expression, such as insulinomas, neuroblastoma, pheochromocytoma, medullary thyroid carcinoma, and paragangliomas [22].

Additionally, it must be taken into consideration that the physiological distribution of ^18^F-DOPA on pancreas limits its ability to evaluate panNENs. However, to avoid masking of a potential lesion, carbidopa premedication may be attempted to reduce the physiological pancreatic uptake and to increase the tumor’s uptake, making the lesions more easily detectable [22,80,81]. Helali M. and colleagues investigated the use of ^18^F-DOPA PET/CT with carbidopa premedication in 20 patients with non-functioning panNENs. According to this study, normal pancreatic parenchyma was only faintly visible in all patients, confirming the effective inhibitory influence of carbidopa premedication on physiological uptake of ^18^F-DOPA. The reported sensitivity in identifying nodal localizations, distant metastases, and primary panNEN was 90%, 81%, and 100%, respectively [82]. Additionally, in the post-therapeutic setting, ^18^F-DOPA may aid in the detection of new lesions [5]. Veenstra et al. compared ^18^F-DOPA and ^68^Ga-DOTA-TOC before and after PRRT, concluding that ^68^Ga-DOTA-TOC-negative lesions and ^18^F-DOPA-positive ones may also respond to PRRT [83].

### 2.5. New Radiopharmaceuticals

In addition to SSTR agonists, antagonist radiopharmaceuticals have been developed for the study of well-differentiated NET patients. These radiopharmaceuticals were evaluated in preclinical studies and compared with agonists. After binding to receptors, they are not internalized and recognize a greater number of binding sites both in vivo and in vitro, thus showing high tumor uptake and retention [84,85].

The most promising radiopharmaceutical is ^68^Ga-NODAGA-JR11 (^68^Ga-OPS202), an SSTR antagonist that exhibits high affinity for SSTR2 and an excellent biodistribution profile with high target/background ratio, thus allowing the detection of high number of lesions [86]. Dosimetric studies showed that the highest tumor contrast was observed 1 h after injection and calculated that the effective dose is similar to the agonists [86]. One of the most important differences in biodistribution, as compared with agonists, is the higher tumor-to-background ratio due to lower absorption in normal tissues, such as the liver, spleen, and gastrointestinal tract. In a study of 12 patients with GEP-NEN, ^68^Ga-OPS202 proved to be more sensitive than ^68^Ga-DOTA-TOC (88–94% vs. 59%), thanks to a higher detection rate of liver metastases, the most common secondary lesions in panNENs. By contrast, the detection rate of lymph node metastases was comparable between the two radiopharmaceuticals [87].

Another study focused on the role of another SSTR2 antagonist, ^68^Ga-DOTA-JR11, in patients with metastatic NEN. This radiopharmaceutical showed a higher detection rate for liver and spleen lesions than agonists, a comparable detection rate for primary tumors and lymph nodes, but a lower detection rate for bone lesions [88]. ^68^Ga-NODAGA-JR11 and ^68^Ga-DOTA-JR11 differ from the chelator used to label JR11. The chelator NODAGA affects SRRT2 binding affinity less than DOTA [85,89]. Both antagonists are valid alternatives to SSTR agonists in patients with predominant hepatic metastatic NENs [90].

The potential therapeutic role of somatostatin antagonists is currently under investigation. In a phase I study of well-differentiated NETs, high doses of ^177^Lu-satoreotide-tetraxetane were administered to nine patients with panNET. It was shown that NETs can be irradiated with a favorable tumor-to-normal-organ dose ratio. However, a high hematotoxicity, greater than expected, was observed with the same doses of SSTR2 agonists [91].

Also worth of mentioning are the new fluorinated somatostatin analogues currently under investigation. Advantages include the longer half-life of ^18^F (110 min) compared with ^68^Ga (68 min) and the shorter positron range of ^18^F (0.6 mm vs. 3.5 mm), which allows high-resolution images. Among these, ^18^F-SiFAlin-TATE has emerged as a promising radiopharmaceutical for PET/CT imaging of NENs, featuring high tumor uptake, a kit-labelling method, and excellent image quality. In a study of 13 patients with panNET G1/G2 who were also investigated with ^68^Ga-DOTA-TOC, the biodistribution of ^18^F-SiFAlin-TATE, 1 h after injection was very similar to that observed with ^68^Ga- DOTA-TOC [92]. ^18^F-AlF- NOTE-octreotide was found to be safe and well-tolerated, with favorable dosimetry, biodistribution, and tumor targeting, as well as a lesion detection rate and tumor-to-background ratio comparable to the current clinical standard with ^68^Ga-DOTA-TATE [93].

## 3. Conventional Radiological Imaging

In the management of panNENs, conventional imaging plays a key role in lesion detection, characterization, staging, surgical planning, and monitoring of the course of disease [27,52]. With the term of conventional imaging, radiologists refer to contrast-enhanced computed tomography (CT) and magnetic resonance imaging (MRI), which are the two main methods of imaging routinely used [27,52]. The performance of transabdominal ultrasound (US) is limited, due to low accuracy in panNEN detection and characterization. However, it should be considered to study the liver parenchyma in the case of unavailability of CT and MRI [94].

The first method of imaging in panNENs workup is contrast-enhanced CT, taking into account the large availability, reproducibility, and high level of sensitivity [1,2]. Concerning panNENs’ detection, contrast-enhanced CT yields a sensitivity ranging from 61 to 93%; these data are strictly linked to tumor size and vascularization [52,95]. Indeed, the performance of CT in case of iso-enhancement and small panNENs (diameter < 10 mm) is poor, and approximately 68% of small panNENs could be missed [96]. The use of contrast medium with multiphase acquisition is essential in neuroendocrine tumors, which usually present homogeneous hypervascularity, appear as well-circumscribed lesions, and do not infiltrate the main pancreatic duct. These characteristics are recognized worldwide as typical neuroendocrine CT findings, and correlated mostly with good prognosis and low tumor grading [97]. Nevertheless, the performance of contrast-enhanced CT failed in tumor characterization in the case of atypical panNENs frequently linked to high tumor grade and poor prognosis [97,98]. The group of Kim et al. [98] demonstrated that the differentiation of panNENs, with uncommon CT findings and high tumor grade, from pancreatic adenocarcinomas could be difficult, and the clinicians must be aware at the time of diagnosis to reduce the misdiagnoses and assume the wrong therapeutic approach. Furthermore, the performance of abdominal CT in the assessment of liver metastases is correlated with the lesion dimension and vascularity; overall, the CT detecting rate is lower in comparison with MRI and functional imaging [99]. In such a scenario, the main clinical risk is to under-stage, then the staging workflow should be always implemented with the use of abdomen MRI.

Multiparametric MRI represents a valid imaging method to cover the main gaps of contrast-enhanced CT, especially in tumor detection and in M-staging; these aspects are mandatory to choose the best therapeutic options and for surgical planning: Figure 4 [27,52]. In particular, the detection rate of morphological sequences in small panNENs is similar to contrast enhanced CT, but the implementation of diffusion-weighted images improved the sensitivity in both panNENs detection and in assessment of liver metastases [100]. Furthermore, the use of hepatobiliary phase with Gd-EOB-DTPA should always be considered to have a complete evaluation of liver involvement, and by using the combined set of diffusion-weighted images and hepatobiliary phase, allows high value of sensitivity and specificity to be achieved in the step of diagnosis and during follow-up [101]. Recently, MRI was investigated to identify high-risk patients, evaluating the features correlated with tumor grading and early progressive disease after surgery. Indeed, Canellas R. et al. [102] demonstrated that MRI could be used to predict poor prognosis in the case of tumor size major of 2 cm, non-bright signal of lesions on T2, in the presence of both duct dilatation and restricted diffusion.

To sum up, conventional imaging is essential for an appropriate work-up and the option to integrate several imaging methods should be always considered to avoid patient under-staging and to choose the most appropriate therapeutic strategy.

## 4. Radiomics

Radiomics is a non-invasive quantitative imaging approach, able to reflect tumor microarchitecture, lesion heterogeneity, and environmental by using the numeric features extracted from medical images, for the chance to have an overall assessment of tumor aggressiveness: Figure 5 [103]. In the new era of personalized medicine, quantitative imaging is becoming central, providing an objective tool to the oncologists to stratify the patients according to tumor behavior [104]. To date, several studies investigated the role of radiomics in management of oncologic patients, from diagnosis to risk stratification [103,105]. Focusing on panNENs, consistent radiomic studies are available in the literature to explore the performance of quantitative imaging in NETs workup, aiming to provide information on tumor characterization degree differentiation and predicting response to target therapy [106,107,108,109,110,111].

Tumor characterization represents an actual challenge for conventional imaging, especially in the case of atypical or unknown NENs, and, in that context, a quantitative approach was proposed by He M. et al. [110]. They tested the radiomic approach to differentiate atypical panNENs from pancreatic adenocarcinoma, achieving an AUC of 0.884, while the clinic-radiological model had an AUC of 0.775. They manually segmented the entire volume of each pancreatic NEN; the radiomic features extracted were used to build three different models (radiomic, clinico-radiological, and combined) to distinguish pancreatic adenocarcinoma from panNENs. The best approach transpired to be the radiomic model with the promising AUC of 0.884. Similarly, Martini I. et al. [112] tested the differences of quantitative parameters extracted from liver metastases in gastrointestinal NENs and panNENs, and they demonstrated significant differences between the two patient groups.

Furthermore, consistent results were reached in differentiation of panNENs tumor grading by performing a volumetric segmentation, extracting radiomic features from abdominal CT and building radiomic and combined scores, as well as merging clinical parameters [108,109,110]. The major results were achieved by the group of Bian Y. et al. [108], who performed a CT-based radiomic score to differentiate G1 from G2 in nonfunctioning panNENs, yielding an AUC of 0.86, sensitivity of 94%, and specificity of 63%. Similar results were shown by the multicentric study of Gu D. et al. [109], who built a combined nomogram (clinical plus selected radiomic features) to differentiate panNENs G1 from G2/G3. The AUC was performant with values of 0.974 and 0.902, in the training and validation cohort, respectively.

Recently, radiomics was also tested in NENs as an imaging tool to predict response to therapy in patients with metastatic NENs with progressive disease who could benefit of treatment with Everolimus [111]. Radiomic approach transpired to be promising in differentiating patients’ responders from non-responders before starting the therapy, and the rad score outperformed the clinical score, demonstrating that quantitative imaging could play a key role in the personalized medicine.

### Radiomics and Nuclear Medicine

The long tradition of nuclear medicine in exploring the “function” of different biological processes and in providing qualitative and quantitative information, makes this specialty particularly suitable to radiomic approaches. Indeed, in the past decade, the nuclear medicine community paid a great deal attention to the possibility to better characterize tumor heterogeneity by means of these novel opportunities, as demonstrated by the increasing number of published papers focusing on radiomics and SPECT or PET.

Tumor heterogeneity is related to several factors, for example, necrosis, cellular proliferation, angiogenesis and micro-vessels local density, and hypoxia. Increased tumor heterogeneity is reported to be a negative prognostic marker in many cancers and to be associated with a worst response to treatment [113,114].

In this scenario, radiomics could be able not only to better characterize tumoral heterogeneity but also to identify patterns and changes after therapy that cannot be appreciated by the human eye, thus providing crucial prognostic information.

In particular, it is well known that the clinical aggressiveness of NEN and the success of therapies strictly rely on the expression of SSRs on the tumor surface. As previously mentioned, ^18^F-FDG is a valid alternative to ^68^Ga-SSA PET/CT in patients with poorly differentiated NENs. Nevertheless, this de-differentiation may occur in later stages of the disease and, therefore, it would be desirable to apply to novel approaches that are able to promptly provide prognostic information. Although very few papers focused on radiomics and NM in NENs exist and are referred to mainly in case reports [115] or very limited populations and heterogeneous cohorts, they achieved interesting results.

Werner et al. retrospectively enrolled 31 G1/G2 panNET patients studied with ^68^Ga-SSA PET/CT prior to PRRT with ^177^Lu-DOTATATE. In addition to conventional PET parameters, such as SUVmax, SUVmean, and MTV, textural features were also extracted from PET images, and their predictive ability for OS was analyzed [116]. All the clinical and PET parameters failed to predict the response to PRRT, whereas entropy was significantly associated to OS (high entropy is predictive of longer survival, low entropy is predictive of worst outcome) in the entire group as well as in the sub-population of G2 panNET patients. Confirming the results previously achieved in another paper published from the same group [117], the authors concluded that textural features analysis, reflecting the heterogeneity of SSTR expression at baseline PET/CT study, may be very useful for risk stratification of these patients. With similar purposes, Wetz et al. analyzed the predictive role of asphericity in 20 GEP-NET patient candidates to PRRT by using ^111^In-DTPA-octreotide SPECT/CT scintigraphy [118]. Asphericity is a quantitative approach to evaluate the spatial heterogeneity of SSRs in tumoral lesions. In this study, this novel parameter was compared with traditional visual and semi-quantitative evaluation, represented by Krenning score and metastasis to liver uptake ratio (M/L ratio) in terms of response prediction to PRRT. Compared with conventional interpretation criteria, the authors found that higher asphericity was significantly associated to worst outcome and poorer response to PRRT, thus better reflecting the spatial variations of SSTRs in primary tumor and its distant metastases. They, therefore, concluded that asphericity could be a reliable parameter for predicting the response to PRRT in centers where PET/CT is not available [118]. More recently, another group evaluated the coefficient of variation with ^68^Ga-peptides PETC/CT, which indicates the percent variability of SUVmean within tumor volume, thus reflecting the heterogeneity in STTRs distribution [119]. Interestingly they found that all malignant lesions had up to 4-fold higher coefficient of variation compared with normal tissues. Compared with primary tumor, bone metastases had a significantly higher coefficient of variation, whereas liver metastases had significantly lower coefficient of variation. This heterogeneous distribution of SSTRs within different lesions of the same patient should be taken into consideration when scheduling a therapy with hot or cold SSAs because different lesions could show different response to therapy [28]. Onner et al. investigated the predictive value of skewness and kurtosis on pre-treatment ^68^Ga-peptides PET/CT, concluding that these two textural features were able to differentiate responder from non-responder lesions to PRRT, with skewness and kurtosis being significantly higher in lesions that did not respond to treatment [120].

Other groups focused on treatment-related changes after PRRT [121,122]. In a small and heterogeneous series of NEN patients treated with different approaches, Weber et al. explored the possible role of textural analysis of ^68^Ga-DOTATOC PET/MRI in identifying SSTRs fluctuation after therapy [30]. They found a trend in entropy decrease on ADC maps in responder patients compared with non-responders, but none of the radiomic or PET parameters analyzed was able to predict treatment response to PRRT [121].

Conversely, in a more recent retrospective study, another group analyzed 324 SSTR-positive lesions from 38 GEP-NEN patients studied with ^68^Ga-DOTATOC PET/CT prior to and after PRRT, extracting 65 PET features from each lesion [122]. In this study, HISTO_skewness and HISTO_kurtosis were able to predict treatment response to PRRT, both being significantly higher in non-responder lesions than in responder lesions, regardless of their origin and location. PET parameters and clinical parameters alone were not able to provide such prognostic information [122].

Despite the very promising role of radiomics in many clinical contexts, it has consistent drawbacks, and the application in clinical routine is limited and not standardized. Reproducibility, lack of validation, and poor availability are the main limitations, which could be overcome by testing radiomics in multicenter and prospective studies. Quantitative imaging is expected to be a supporting tool for the clinicians and not a replacement of the conventional clinical and radiological approaches, with the goal to standardize a structured workflow for the patients and to reduce the subjective evaluation.

## 5. Conclusions and Future Perspectives

The management of NENs has radically changed in the past decades thanks to the improvement of diagnostic modalities and therapeutic strategies. Radiology, nuclear medicine and, more recently, radiomics offer a wide plethora of approaches able to provide crucial information for staging, risk stratification, follow-up, and therapy monitoring. The combination of these modalities and the discussion of each single case within a multidisciplinary team are crucial aspects for ensuring an optimal management of the patients and for planning the most appropriate therapeutic strategy. In the era of “personalized therapy” and “theragnostic” approaches, there is, indeed, the increasing clinical need to better understand tumor biology and to better characterize its heterogeneity, thus ensuring the best outcome for these patients. Novel radiopharmaceuticals are currently under investigation, aiming to provide more accurate tools for therapy planning and evaluation. These new imaging strategies, together with radiomic approaches and novel therapies, such as immunotherapies, will contribute in the near future to changing the way to approach to NENs, thus further improving patient’s management.

## Figures and Tables

**Figure 1 jcm-11-06836-f001:**
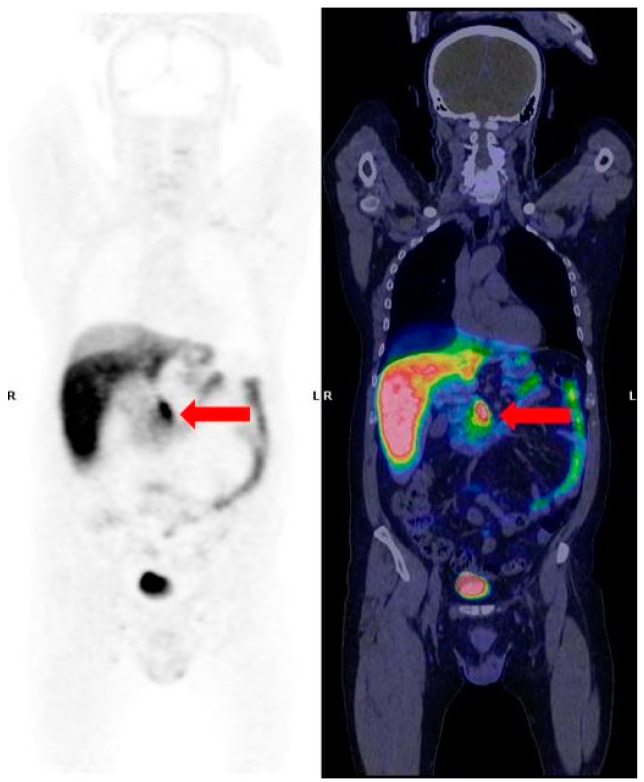
^68^Ga-DOTA-TOC PET/CT coronal images. The red arrows show a very common finding: a physiological diffuse uptake in the pancreatic head/uncinate process.

**Figure 2 jcm-11-06836-f002:**
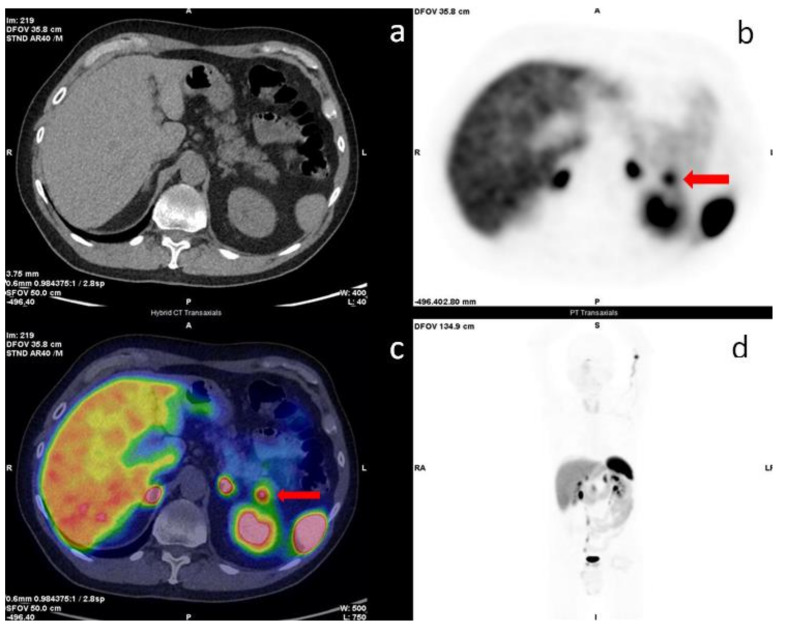
^68^Ga-DOTA-TOC PET/CT transaxial images. Low-dose CT: (**a**), PET: (**b**), image-fused PET/CT: (**c**), and maximum intensity projection (MIP) (**d**). The red arrows in images (**b**,**c**) show a focal uptake in the pancreatic tail while restaging patient for small bowel NET (G2, Ki67 3%). The morphological imaging was not decisive for the diagnosis. Cytology revealed accessory intrapancreatic spleen.

**Figure 3 jcm-11-06836-f003:**
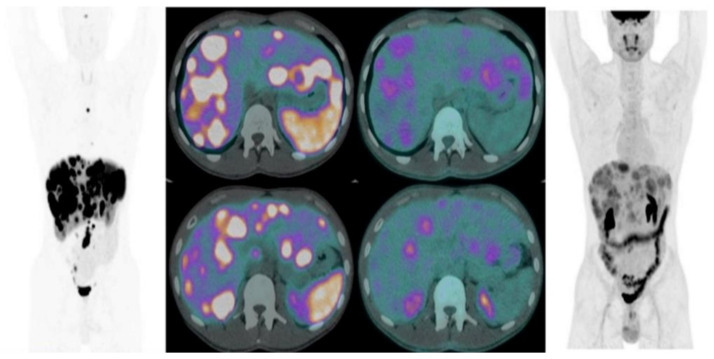
Combined imaging ^68^Ga-DOTA-SSAs-PET/CT and ^18^FDG-PET/CT in G2 small bowel NET showing heterogeneity of the tumor.

**Figure 4 jcm-11-06836-f004:**
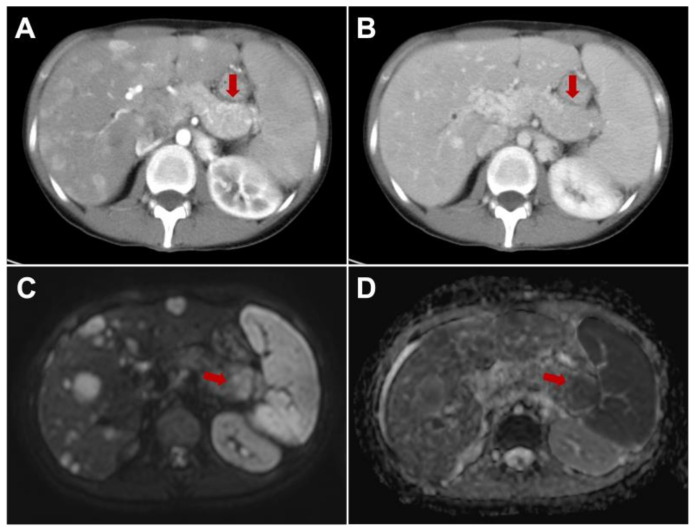
A 35-year-old female with atypical metastatic pancreatic neuroendocrine neoplasm of the tail, difficult to identify in contrast-enhanced CT due to the iso-enhancement in arterial ((**A**), arrow) and portal ((**B**), arrow) phase. It was detected by MRI by using diffusion-weighted images, showing hyperintensity on the DWI ((**C**), arrow) and low signal intensity on the ADC map ((**D**), arrow).

**Figure 5 jcm-11-06836-f005:**
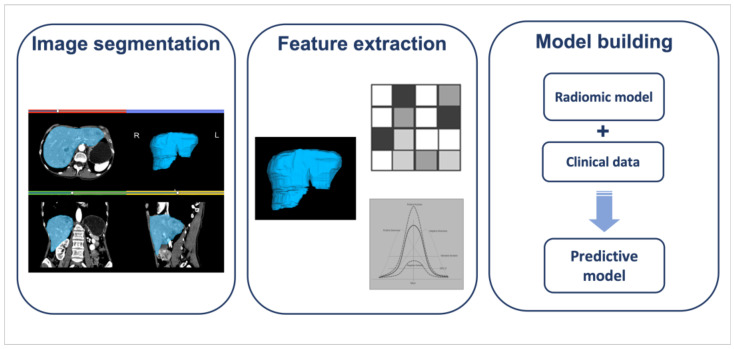
Graphical representation of radiomic workflow from liver segmentation, in panNEN with liver metastases, to build a predictive model.

## Data Availability

Not applicable.

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
