# Peer review of "Nuclear Medicine and Radiological Imaging of Pancreatic Neuroendocrine Neoplasms: A Multidisciplinary Update"

_jcm, 2022, doi:10.3390/jcm11226836_

Round 1

Reviewer 1 Report

This review article by Prosperi et al. might be valuable in understanding the existing evidence in nuclear medicine and radiomics. Generally, the content is well-written.  There might be some points to modify (minor revision).   1. There are several redundant descriptions. For example, in Page2 line 73, "As it is known" is unnecessary. In addition, the sentence " this has been considered a limiting factor" can be modified as " this has been a limiting factor". Authors should consider descriptions thoroughly.   2. In page 3 line 108-110, please clarify the definition of "malignant" insulinoma. In addition, in reference [35] , only 2 cases of metastatic insulinoma cases were included. Is there any evidence showing the expression pattern of SSTR in malignant insulinoma?   3. In page3 line 111-145, authors show possibilities of false positives. This is valuable information, but the description is too redundant. For example, in line 128, authors begin the paragraph with "Rarely", in line 133, they write " Although it is a very rare finding".  The content is repeating. Please consider making the description simple and easy to understand.    4. In page4 line 178, the meaning of the sentence " the decision to perform 18F-FDG 178 scan in well-differentiated tumors, especially G1 and low-grade G2 (Ki-67< 10%) is 179 mainly referred to a multidisciplinary evaluation." might be slightly difficult to understand.   5. Please summarize the enumeration of previous reports about the combined use of  18F-FDG 181 and 68Ga-SSA PET/CT(line 185-). In other words, please clarify the benefit of combined use of these modalities as well as clarifying the role of each modality.   6. How about discussing the utility of MTV and TLGs in the section of 18F-FDG-PET/CT?

Author Response

file enclosed

Reviewer 2 Report

 Prosperi et al. reviewed the role of conventional imaging techniques and new perspectives in nuclear medicine for the diagnosis and management of pancreatic NETs. This is a well written manuscript by experienced authors in the field. I have some minor suggestions:

1) Page 2, lines 88-94: The authors should briely discuss the factors affecting preference among various Ga68 dota-peptides although they emphasize that there are no significant difference clinically among these radiopharmaceuticals.

2) Page 3, line 105: Regarding the utility of SSR based imaging in insulinoma, the authors report two different sensitivities (25-30% in line 106) and 85% in line 107). Please clarify this issue.

Does the authors mean “25-30% SSR positivity” of instead of “sensitivity” ? The same issue is mentioned correctly in page 6, line 268.

3) Page 3, line 116: It would be better if the authors emphasize that increased activity of uncinate process in SSR imaging should be confirmed by convetional imaging methods such as CT before diagnosing as NEN of pancreas or asan  another pathologic imaging.

4) Page 6, lines 296-297:  “…………….role of the 18F-DOPA PET/CT in the management of patients with entero-pancreatic NENs is …………”. It should be “imaging” insteat of “manegement”.

5) Page 7, lines 336-337: “The most promising radiopharmaceutical is 68Ga-NODAGA-JR11 (68Ga-OPS202), an 336 SSTR2 receptor ligand that exhibits …”. Since 68Ga-OPS202 is a SSR antagonist, it should not be considered as SSR ligand. Please improve the sentence.

6) Page 9, lines 434-440: The authors give some AUC numbers derived from CT radiomics for differentiating pancreatic NEN from non-neuroendocrine neoplasms of pancreas. It would be better for the reader to give a brief explanation for radiomics obtained from CT and the meaning of AUC values (how they derived/calculated). Otherwise, those numbers will not be helpful to some of the readers.

Author Response

file enclosed
